# ESG Reporting and Metrics: From Double Materiality to Key Performance Indicators

Christian Nielsen 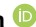

Department of Computer Science and Engineering, University of Bologna, 40126 Bologna, Italy;
christian.nielsen@unibo.it

**Abstract:** This article conceptualises the link between firms' value chains and distribution networks and the requirements for double-materiality assessments in contemporary reporting regulations worldwide. The new European Sustainability Reporting Standards (ESRS) and the standards for sustainability reporting issued by the International Sustainability Standards Board (ISSB), called IFRS S1 and IFRS S2, require companies to report their own direct (scope 1) and indirect (scope 2) greenhouse gas (GHG) emissions as well as GHG emissions in their value chains and distribution networks (both scope 3). However, GHG emissions comprise just one dimension of information that is relevant to understand when assessing, managing and reporting the footprints and impacts of a firm and are, therefore, only a fraction of the key performance indicators (KPIs) related to ESG that should be disclosed. Through a case study, this article demonstrates the connection between a due diligence analysis of a firm's value chains and distribution networks; an analysis of the competitive parameters of its business model; the identified impacts, risks and opportunities; and the double-materiality perspective. The double-materiality perspective prioritises actions based on probability and significance, creating a natural space to identify KPIs. The implication of this study is that firms can be assisted in identifying relevant KPIs based on double-materiality assessments aided by applying the REGS model because it guides firms in choosing the most relevant KPIs.

**Keywords:** ESG; sustainability reporting; due diligence; impact; footprint; double materiality; key performance indicators





## 1. Introduction

A contemporary understanding of corporate footprints that encompasses the needs of internal and external stakeholders, the impacts we make, and that which impacts us is needed. Nielsen [1] argues the case for re-formulating the current accountability debate concerning corporate social responsibility reporting, Environmental, Social and Governance (ESG) information, and sustainability reporting in its many varying formats so that it focuses on the most significant impacts and connects with firm performance [2].

Due to the ever-increasing awareness of ethical and sustainable business practices, organisations are pressured to connect their business practices and strategies to impacts on the environment, society and stakeholders they interact with [3]. The European Sustainability Reporting Standards [4] specify the information an organisation should disclose about such material impacts, as well as the risks and opportunities concerning environmental, social and governance sustainability matters the company faces. According to the ESRS, a sustainability report must describe "the key elements of the undertaking's general strategy that relate to or affect *sustainability matters* and the key elements of the undertaking's *business model* and *value chain* to provide an understanding of its exposure to *impacts, risks* and *opportunities* and where they originate" [5].

The current focus of governments and supra-national bodies emphasises the importance of creating transparency. One example is the OECD's Financing SMEs for Sustainability [6] platform. Assisting SMEs in becoming transparent regarding their impact is

essential in ensuring accountability and sustainability in global value chains. In a recent contribution, Roslender and Nielsen [7] find that one of the fundamental mechanisms that can hold businesses accountable for their sustainability goals is the interest and power of customers. However, they also conclude that the importance of customers as stakeholders is under-emphasised in current corporate reporting frameworks. They risk becoming merely an elaborate marketing material that is better described as "doing-good reporting". In donating towards and investing in the United Nations' Sustainable Development Goals, doing good is a potential addition to impacts. Still, it is also attested that the focus on sustainability should be based on arguments about a company's impacts on its surroundings stemming from its core activities to provide concise information on the business's impacts [1].

The present article answers the following research question: how can firms create a link between their business model and the ESG KPIs that document their double-materiality statement?

In today's globally oriented economy, companies seldom operate in isolation; they often work in complex global value chains and ecosystems, interacting with many other organisations and individuals. Therefore, an integral part of a value chain and due diligence analysis is concerned with describing how a focal firm affects and is affected by its nearest business partners and how value creation, value capture and value destruction are dispersed among the companies in the value chain. Value destruction could, for example, negatively affect a value chain's climate through excessive greenhouse gas (GHG) emissions. In a supply chain, every company needs to focus on how value creation and destruction are dispersed across stakeholders. If it is difficult to project these values across the value chain—and often it is—then what would be necessary to improve this type of transparency?

In the forthcoming sustainability reporting standards, assessing an organisation's material impacts, risks and opportunities must be validated by a due diligence process to cover the relevant parts of that organisation's value chain. For example, the impact of GHG emissions is categorised according to Scopes 1, 2 and 3 [8]. Scope 1 covers direct emissions from owned or controlled sources. Scope 2 covers indirect emissions from the generation of purchased electricity, steam, heating and cooling consumed by the reporting company. Scope 3 includes all other indirect emissions in the company's value chain. Therefore, when considering value creation and value destruction, it is advisable to consider direct and indirect effects throughout the value chain and more broadly than just GHG emissions. (Figure 1).

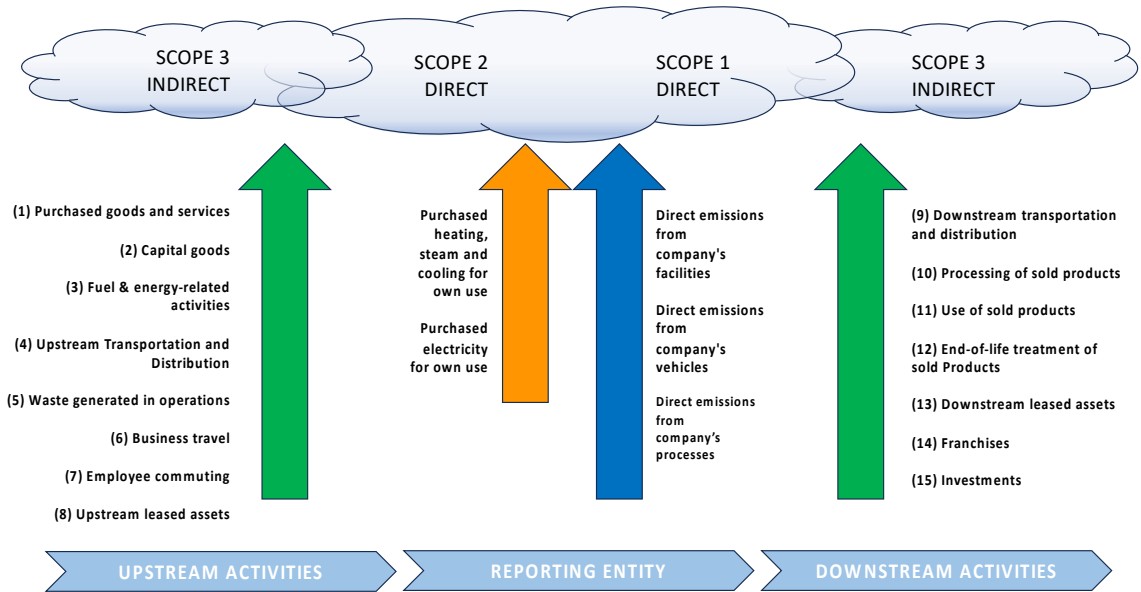

**Figure 1.** Scopes 1, 2 and 3 (own production following [9]).

What is deemed relevant can be assessed based on a business model analysis of a given company and the closeness of the business relationships the organisation has. A due diligence process could involve performing the following steps [10]:

1. Integrating due diligence into policies and management systems;
2. Identifying and assessing adverse human rights and environmental impacts;
3. Preventing, ceasing or minimising actual and potential negative human rights and environmental impacts;
4. Evaluating the effectiveness of measures;
5. Communicating;
6. Providing remediation.

Hence, due diligence takes the analyses performed on value creation, value capture and value destruction a step further and takes action. The crucial link to the remainder of this paper is identifying measures that assist due diligence. This is exemplified by a case study using qualitative research methodologies, as explained in Section 4.

If your performance measures do not reflect your business model, you are probably not getting what you have bargained for. In a similar vein, it can be argued that if a company's chosen KPIs do not enable it to assess the effectiveness of actions via the due diligence process, it is likely that it will not help managers understand whether the company is performing well or not.

## 2. Background: Identifying Double Materiality

More research needs to be conducted on actual double-materiality reporting practices. One recent study [11] highlights that in its infancy, the notion of double materiality has led to a relatively wide variety in both double-materiality assessments and adoption disclosures, as well as related criticalities. Describing how an organisation affects people, the planet and society while creating profits is at the core of sustainability reporting. This is denoted as having an impact. The ESRS states that when describing the process to identify material impacts, risks and opportunities, a company should disclose all relevant criteria used in the process. An example of this clarity is found in the ESRS:

"ESRS 2 SBM-2 requires the undertaking to provide an understanding of if and how it considers whether its strategy and business model plays a role in creating, exacerbating or (conversely) mitigating significant material impacts on *consumers and end-users,* and whether and how the business model and strategy are adapted to address such material impacts" [12].

The ESRS and other forthcoming international sustainability reporting standards use the expression "materiality of the impacts". This means that organisations must evaluate the extent of their impacts or footprints, for example, by starting with understanding the business model. This means that organisations must focus on areas where impacts, risks and opportunities are likely to arise based on the nature of their activities, business relationships, geographies or other factors. Double materiality means that a company has to evaluate its footprint on the environment and society on the one hand and how environmental and societal factors influence the organisation on the other. The former is called impact materiality, and the latter is financial materiality. From the ESRS, we derive the following definitions:

**Impact materiality**

A *sustainability matter* is "material from an impact perspective when it pertains to the undertaking's material actual or potential, positive or negative *impacts* on people or the environment over the short-, medium- or long-term.

Impact materiality considers a company's most significant impacts outwards for the most significant stakeholders and should be presented in the sustainability report primarily for external non-financial stakeholders.

**Financial materiality**

A sustainability matter generates *risks* or *opportunities* that have a material influence, or could reasonably be expected to have a material influence, on the undertaking's development, financial position, financial performance, cash flows, access to finance or cost of capital over the short-, medium- or long-term.

Financial materiality considers a company's most significant impacts inwards and should be presented in the annual report. It is specifically intended for investors, lenders and other creditors.

A process for determining material topics is depicted in Figure 2. The first phase examines the context, the strategy and the business model. It is recommended that companies engage with their stakeholders in such discussions. In determining the most significant impacts, a company must look beyond its limits and include stakeholders and the value chain it affects, as is argued above regarding due diligence. Therefore, a natural starting point for such an analysis is to specify how a company creates value, i.e., the business model. This also determines the degree and manner of collaboration outside the immediate organisation. The information derived here is equally relevant to managerial decision making and external communication. In the second phase, the impacts are prioritised for reporting purposes. Here, the Global Reporting Initiative (GRI) [13] proposes double-checking the materiality of topics and prioritising with potential information users and experts that can verify their significance.

A series of analyses can be conducted to become more precise on the utility of products and services from the perspective of customers, value creation and value destruction. This could include describing the business model configuration, the type of utility it aims to create for the company's customers and users (the value propositions), and the primary value-creating activities. A helpful question is, "What are the particular value drivers of our business model?" Conducting such a business model analysis makes it possible to identify the company's impact and footprints on society, individuals and the environment. A helpful starting point is to contemplate areas where the company's value creation depends on the use of resources, which could potentially cause value destruction of natural resources or materials and components.

Clarifying a firm's value creation and types of value destruction and how the processes and activities affect other companies and the environment makes it possible to create a clear and transparent illustration of impacts. An effective way to communicate this is to use the Future-Fit framework [14] or the six capitals from the Integrated Reporting Framework [15]. These two frameworks are among the most widespread for illustrating the impacts and footprints of companies, but they do not distinguish between the two sides of double materiality.

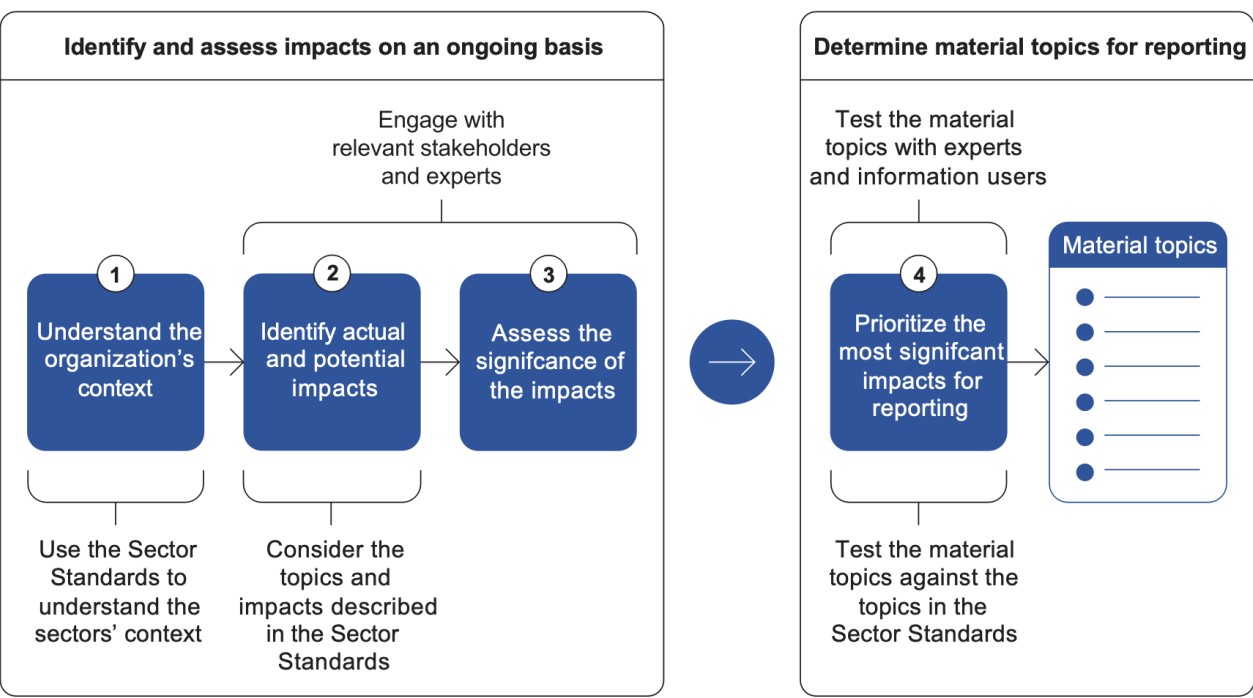

**Figure 2.** Identifying impacts and determining their materiality [13].

The Future-Fit Business Benchmark tool is a free online strategic management tool that can assist companies in assessing and managing the impacts of their activities environmentally, socially and financially. The tool lists 23 Break-Even Goals across eight categories (energy, water, natural resources, pollution, waste [16], physical presence and human drivers of Future-Fit pursuits). Not all classes are relevant for a given company. Which goals should be focused on can be determined by considering the business model and the types of value creation, as articulated above. Similarly, Integrated Reporting identifies categories that could be relevant, such as natural capital, physical capital, human capital, intellectual capital, social capital and financial capital. There are three overall categories according to which we can analyse footprints: (1) the footprint of the inputs to a company's production, (2) the footprint of the processing in the company's production, including value delivery and effects on the workforce, and (3) the footprint of product use, including costs of circularity, reuse, and discarded materials. When illustrating a footprint, companies should clarify the following:

- Why do they choose to report this particular footprint?
- What are the key stakeholders affected by a business regarding this footprint?
- Who else would a company like to affect and why? A company can choose to invest or donate here. For example, which SDG goals would a company like to invest in that are not directly affected by its current operations or products?

When an organisation has identified its impact materiality and financial impacts, it needs to prioritise the "most significant impacts" and validate the potential effects. One way to handle this analysis is to create an overview using a double-materiality matrix like the one depicted below, inspired by the SASB and GRI and shown in Figure 3. A double-materiality matrix and explanatory text are integral to a sustainability report. They should identify which impacts are of most significant concern to the organisation, both outwards and inwards and in combination.

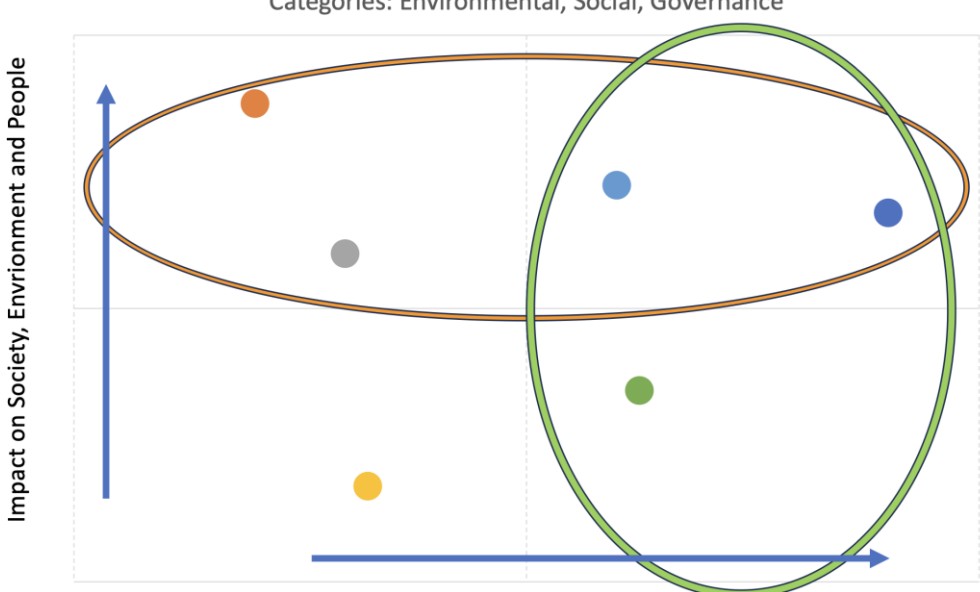

**Figure 3.** Double-materiality matrix (own production, inspired by the SASB and GRI).

## 3. Identifying ESG KPIs

A core part of creating reliability of sustainability reporting and working towards greater comparability and transparency for financial stakeholders is to have such reports assured and audited. The forthcoming ISSA 5000 auditing standard [17] will provide a basis for sustainability auditing and assurance. Financial reporting provides criteria that auditors can use to evaluate a company. For external stakeholders, it is essential that reporting accurately represents (faithful representation) the company's condition and that the information is comparable across time and other entities. The International Auditing and Assurance Standards Board's exposure draft, which proposes General Requirements for Sustainability Assurance Engagements [17], expands upon these criteria along five dimensions:

72 (c) Evaluate whether the criteria exhibit the following characteristics: (Ref: Para. A172-A178)

(i)     Relevance (Ref: Para. A179-A180);
(ii)    Completeness (Ref: Para. A181);
(iii)   Reliability (Ref: Para. A182);
(iv)   Neutrality (Ref: Para. A183-A184);
(v)    Understandability (Ref: Para. A185).

On page 57, the ISSA 5000 [17] states that sustainability information relates to "information about sustainability matters and may cover several topics and aspects of those topics". According to the ISSA 5000 [17], the topics include the following:

a.     Climate, including emissions.
b.     Energy, such as type of energy and consumption.
c.     Water and effluents, such as water consumption and water discharge.
d.     Biodiversity, such as impacts on biodiversity or habitats that are protected and restored.
e.     Labour practices include diversity, equal opportunity, training and education, and occupational health and safety.
f.      Human rights and community relations, such as local community engagement, impact assessments and development programmes.
g.     Customer health and safety.

h.   Economic impacts include government assistance, tax strategy, anti-competitive behaviour, anti-corruption and market presence.

These topics are then to be analysed across a series of key aspects that represent how the topics are anchored in managerial and governance processes:

i.    Governance.
ii.   Strategy and business model.
iii.  Risks and opportunities.
iv.   Risk management or mitigation.
v.    Innovation to address risks and opportunities.
vi.   Metrics and key performance indicators.
vii.  Targets.
viii. Internal control over monitoring and managing risk.
ix.   Scenario analysis.
x.    Impact analysis, including the magnitude of impacts.

Regarding measuring financial materiality, the outside-in effect, the International Sustainability Standards Board (ISSB) states the following in the IFRS S1 [18]:

46 An entity shall disclose each sustainability-related risk and opportunity that could reasonably be expected to affect the entity's prospects:

1.   (a) Metrics required by an applicable IFRS Sustainability Disclosure Standard;
2.   (b) Metrics the entity uses to measure and monitor the following:

(i).   Sustainability-related risk or opportunity;
(ii).  Its performance in relation to a sustainability-related risk or opportunity, including progress towards any targets the entity has set, and any targets it is required to meet by law or regulation.

48 Metrics disclosed by an entity, when applying paragraphs 45–46, shall include metrics associated with particular business models, activities or other common features that characterise participation in an industry.

This stresses the importance of creating links between risks and opportunities and how they impact a business, and understanding how these can be measured and the systems in which the data for these KPIs are generated. Therefore, companies must accentuate the connections between these elements and provide longitudinal explanations to increase reliability, trustworthiness, comparability and relevance in sustainability reporting. The data linked to information systems are equally important for internal control and auditing purposes. Hence, new regulations are pressuring companies to build and employ a methodology that links value creation and impacts to KPIs that can be used for managerial purposes and, simultaneously, as a verified basis for sustainability reporting. The ISSA 5000 [17] exposure draft illustrates the connections between sustainability matters, disclosure and how an overview can be created in a process where:

1.   Sustainability matters include the topics and aspects of topics;
2.   These are measured against applicable criteria;
3.   This leads to sustainability information about the relevant matters, which is disclosed.

The main principle for choosing to report a KPI should be that it provides relevant insights from a managerial perspective. For a KPI to make sense from a managerial perspective, it needs to inform management about the status and direction of something meaningful to the organisation. This means that the KPI should say something about one of three things:

1.   Whether the organisation is investing appropriate resources into its strategic activities;
2.   Whether the strategic activities are on track in terms of activity level, intensity or efficiency;
3.   Whether the activities are having the anticipated effects.

Multiple international associations, bodies and organisations have proposed lists of KPIs for ESG reporting. Among these are organisations such as the European Commission,

the European Federation of Financial Analysts, the Global Reporting Initiative, the Sustainability Accounting Standards Board, and an array of financial data platform providers such as Bloomberg, consulting firms such as KPMG, and ERP providers such as SAP. Currently, the maturity of some of these KPI lists could be better, and the most trusted benchmark list to date is that of the Global Reporting Initiative. Like the ESRS [2], the GRI [13] distinguishes between general disclosures and sector-specific disclosures.

When looking at the largest firms in the US, they have mostly adopted environmentally oriented business model archetypes (the E) and, to a much lesser extent, archetypes associated with societal and governance-oriented (the S and G) dimensions of business models [19]. This is also reflected in the KPIs suggested by data providers such as Refinitiv Datastream, Bloomberg and the SASB. Naturally, companies start their experimentation with the E of ESG because these types of metrics are calculable in manners closely related to existing accounting frameworks [20].

One way to begin the journey towards a more holistic measurement of sustainability is to use the REGS model [21], which reconciles the CSR and ESG sides of the great sustainability divide. One of the advantages of the REGS model is that it creates common dimensions and boundaries from which it is possible to link CSR and sustainability activities to ESG metrics. In this reconciliation between CSR and ESG, the REGS model identifies four primary ways in which an organisation can pursue sustainability:

1.  Seeking out sustainability as a strategy to stay resilient;
2.  An emission-efficiency perspective on becoming more sustainable;
3.  Ethical behaviour as a sustainability trait;
4.  Sharing and stewardship-based sustainability.

Typically, companies pursue several of these strategies simultaneously, and therefore, the REGS model in Figure 4 is both a platform to identify relevant performance metrics and indicates an organisation's focus on sustainability.

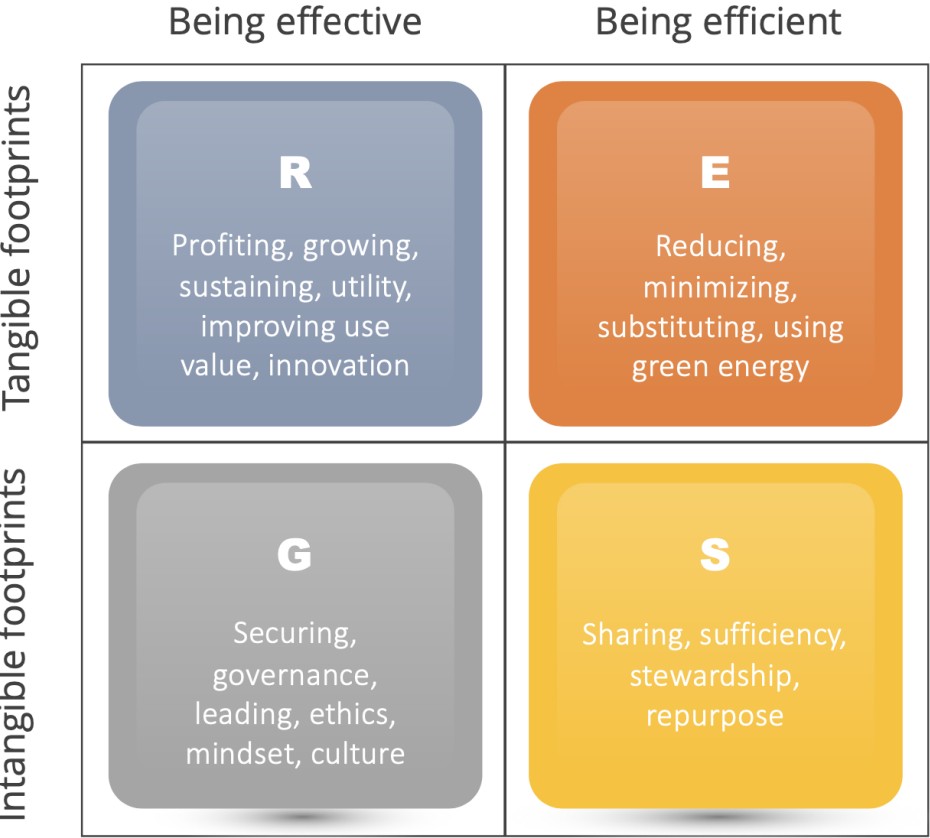

**Figure 4.** The REGS model [21].

## 4. Research Design and Methods

The qualitative case study method was used to explore the link between double materiality and ESG metrics [22]. The data collection included gathering written documents about a company's products and service offerings and semi-structured interviews. Initially, various written documents about Port Esbjerg were collected to enhance the understanding and support a preliminary business model mapping using a Business Model Canvas [23]. Framing the mapping in a Business Model Canvas allowed us to identify the missing pieces and information needs requiring further elaboration [24]. Semi-structured interviews with two internal stakeholders contributed to this, following the themes of the nine building blocks of the Business Model Canvas [25], whereafter the interviewees mapped out Port Esbjerg's current business model using the QUANT tool [26]. The interviews were recorded and analysed in accordance with the interview guide themes by two researchers. These research steps gave us an overview of several fundamental business model changes in Port Esbjerg in the past 20 years. A final data collection exercise was organised as a participative workshop in which participants from the research team and three participants from the case company discussed the prospective business model configurations of the company, related value drivers, and material opportunities and challenges relating to sustainability and digitalisation.

*Introducing the Case of Port Esbjerg*

Port Esbjerg is an international, multimodal transport hub and a critical Scandinavian gateway to the world stage. With over 200 companies employing 10,000 people and generating EUR 2.2 billion in total revenue, the port has been a driving force behind maritime trade and commerce between Denmark and the global community since 1874.

In 1868, the decision to establish Port Esbjerg emerged as a response to Denmark's loss of Altona Harbour following the defeat by Prussia in 1864. With the absence of Altona Harbour, Denmark sought a new western port, and Esbjerg emerged as the ideal location.

Early on, trade with England proved to be a significant business area for the port. Today, the port boasts an extensive route network connecting it to the entirety of Europe.

In 1910, Esbjerg reigned as Denmark's most prominent fishing port, being home to approximately 600 fishing vessels. However, the fishing industry underwent a structural transformation, replacing many small fishing operations with larger players. This shift led to many small fishing companies relocating away from Esbjerg, leaving only a few fishing trawlers operating in the city. These trawlers primarily focused on catching shrimp, crabs and mussels.

By that time, Esbjerg had already cemented its position as Denmark's oil and gas capital. In 1966, the Danish Underground Consortium (DUC) discovered oil traces in the North Sea, and by 1971, the first oil from the Dan field in the North Sea was extracted. The North Sea witnessed a boom in the offshore industry, attracting several major oil and gas companies to establish their presence in Esbjerg, which became the base port for Denmark's offshore industry.

Around the turn of the millennium, a new industry emerged: offshore wind. Companies in Esbjerg were crucial in constructing Horns Rev I, the first large-scale Danish offshore wind farm installed in the North Sea in 2002. Since then, the offshore wind industry has experienced explosive growth, propelling the Port of Esbjerg to the forefront of Europe in handling and shipping wind turbines. Today, more than four fifths of the current offshore wind capacity installed in Europe is shipped from Port Esbjerg.

Port Esbjerg is Denmark's leading roll-on, roll-off port, also known as a RoRo port. Annually, over 4.5 million tons of goods transit through the port.

In 2000, Port Esbjerg transitioned from state ownership to become a municipal self-governing port. Between 2003 and 2014, the port invested approximately EUR 130 million in new facilities and areas to meet the demands of the offshore industry and to lay the foundation for future growth. In 2013, the new port area Oesthavnen opened, spanning 650,000 m$^2$ and being primarily dedicated to assembling, testing and shipping wind tur-

bines. Since then, Oesthavnen has undergone several expansions, and as of autumn 2017, it covers an area of one million m$^2$. Today, Port Esbjerg boasts a total area of 4.5 million m$^2$, making it Denmark's largest port in terms of size.

Port Esbjerg's primary focus is oil and gas and offshore wind. With Denmark's extensive expertise in the wind industry, it is anticipated that the sector will experience continued growth. This growth will inevitably drive the need for infrastructure enhancements in shipping, transportation and Power-to-X facilities.

The competitors to Port Esbjerg are not Danish ports but similar ports in other parts of Northern Europe, such as Hull and Cuxhaven. Port Esbjerg offers various services such as (1) engineering services, including calculations, drawing and tasks on existing vessels; (2) mobilisation, including customising ships for specific wind turbines to be erected; (3) de-mobilisation of ships so that they are ready for new projects; (4) rig services; (5) decommissioning; and (6) stacking.

## 5. Results and Discussion

Port Esbjerg serves three primary industries: gas, oil and wind. Across these three industries, revenues are generated from renting out ground and storage space (40%). Here, customers in these three industries and their suppliers value flexibility and scalability because their activities typically are project-based. Another 40% of revenues come from ship and goods taxes, while the remaining 20% are from crane services and power supply. Port Esbjerg's customers generally value flexible work and contact hours, fast execution and the possibility of hiring dockworkers to complement fixed staff in peak situations. Oil and gas have been the cash cows for Port Esbjerg, but given the recent focus and expertise in wind energy, the harbour is transitioning towards this specialisation. Most of their business, however, is still grounded in oil and gas.

The business model analysis led to the identification of two customer segments across the three industries. These two segments are significantly different across the building blocks of the Business Model Canvas, and the value propositions associated with them are also considerably different.

The first segment entails shipping companies, which are predominantly connected via local agents, and requires a high degree of involvement, interaction and personal assistance. Typically, revenues are made as pay-per-use/service and based on predetermined fixed prices. In some instances, they can perform some forms of self-service. The second segment entails industrial companies, which can be on-site production companies and companies looking to stock goods before shipping them out. This customer segment requires dedicated personal assistance, and there needs to be more room for self-service. Revenues are one-time payments, pay-per-use and ongoing payments based on predetermined fixed prices.

### 5.1. Defining the Business Model and Identifying Its Value Drivers

The first step in defining the business model of Port Esbjerg was to use the Business Model Canvas [23] to organise the data from the interviews and the secondary data as depicted in Figure 5. Combined with this, the output from the QUANT mapping was discussed and confirmed as part of a participative observation process with the respondents from the case company.

The analysis accentuated that three main business model configurations were being applied in the company. These were the Integrator business model, the Leasing business model and the Procurement business model. The Integrator and Procurement business models have their epicentre of value creation in the value configuration [27] part of the Business Model Canvas. This means that the critical value drivers of these two configurations relate to activities, resources and cost structure management. The Leasing business model has its value creation epicentre in the value capture part of the Business Model Canvas because leasing is also a specific type of revenue stream. In addition to leasing revenues, Port Esbjerg gets paid via one-time and ongoing payments in both customer segments.

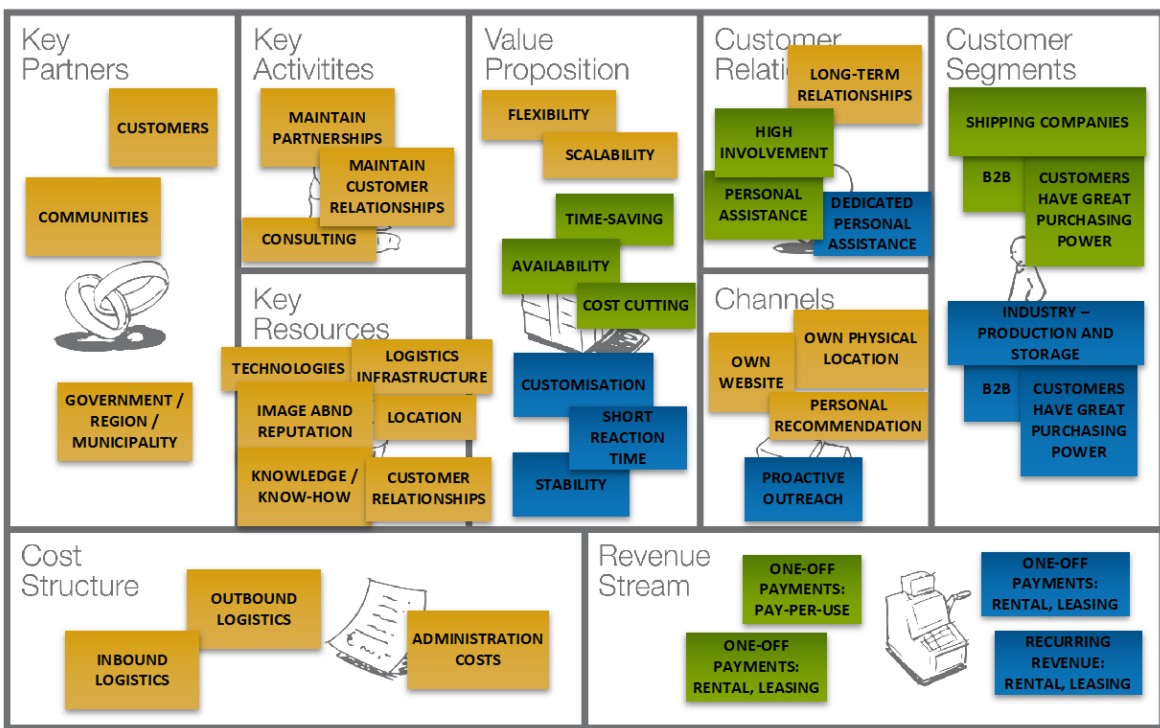

**Figure 5.** Port Esbjerg's Business Model Canvas.

There were slight differences in the value propositions associated with the two customer segments. While it was clear that all customers valued a harbour with flexibility in terms of services and scalability regarding leasing more space or hiring more dockworkers at short notice, the two customer segments differed on a number of other value propositions. The shipping company segment, which typically interacted with the port through several local agents, also valued aspects of the availability of services and that interacting with Port Esbjerg saved time in the overall production. This was also connected to the value proposition of saving costs due to the time aspects and transferring otherwise fixed costs into flexible costs. The second customer segment, including industrial companies, manufacturers and companies assisting with functions related to the stocking and storing of goods, valued the short reaction times from the port. In addition, it was essential for them to know that Port Esbjerg could and would customise their services and deliver consistent and stable service levels.

The value propositions identified the competitive parameters that Port Esbjerg utilised to meet its goals and objectives. Fast execution from an order to delivery was an asset that both customer segments desired. In addition, the port competed for the shipping company segment by being good at reducing customers' costs and being trusted and reliable. Port Esbjerg was competitive in the industrial customer segment due to its customised solutions and the limited availability of its products/services.

It is a pain for the maritime industry to have idle time because its logistics, equipment and ships require significant investments and much coordination. For example, having "idle" rigs is extremely expensive in the oil and gas industries. For the wind turbine industry, the ability to coordinate equipment shipping to the building of wind parks is a significant risk for multinational wind turbine producers and energy companies. Amongst potential business opportunities for ports such as the one in Esbjerg is to act as a value chain service coordinator, facilitating the direction of customers and workflows.

### 5.2. Impact, Footprint and Double Materiality

For Port Esbjerg, green transition is essential in several ways. First, it plays a significant role in the company's future business opportunities. Secondly, it is also a competitive

parameter appearing in the port industry. It is expected to influence the maritime industry's choice of partner ports to a much greater extent in the future. Therefore, a strategic focus on enhancing the port's and its customers' environmental performance will become an essential competitive parameter for Port Esbjerg. The company is also motivated by opportunities to create a CSR-oriented image by becoming certified in different quality and environmental aspects. This is a part of enhancing the possibility of attractive and long-term partnerships because green transition and corporate stewardship are seen as a requirement by customers, business partners, employees and the local community, including the politicians on the company's executive board.

Tightened legal requirements are both an opportunity and a challenge. On the one hand, being at the forefront of environmental and safety regulations in the maritime industry is an opportunity. On the other, such regulations may be at odds with the core operations and possibilities to attract new industrial segments to the port. An example of the latter is the environmental approval of the areas designated to move new wind turbine wings out to the sea and the necessity of increasing the depth in the harbour to receive new jack-up ships, as there must be a minimum depth of 12.5 m for these new larger ships. To be able to deliver on these environmental aspects, the company is challenged by a need for internal competencies to initiate the green transition, and our analyses revealed that changes are generally a challenge for the organisation. Port Esbjerg sees the intensity of technological development in the industry as insignificant. However, digital transformation may be an overlooked strategic opportunity with the spurring importance of Artificial Intelligence, Blockchains, 6G, digital twins and other digital potentials in the Metaverse [28].

The analysis of the company's business model, its material impacts and the financial materiality of its industrial setting identified ten significant material impacts. These were then prioritised according to their materiality degree, as shown in Figure 6 below.

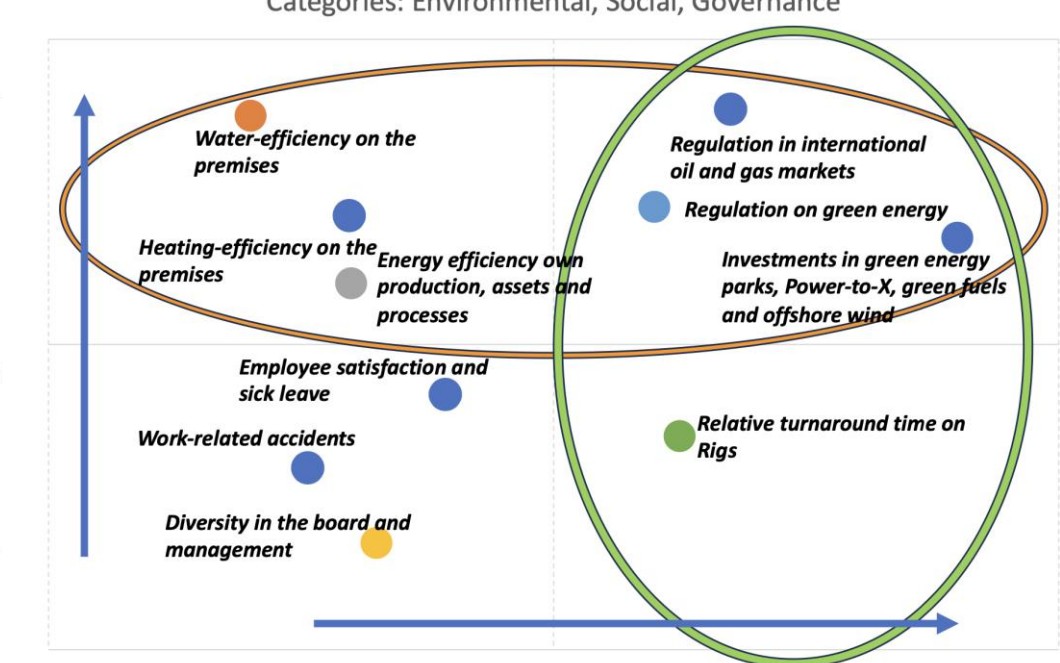

**Figure 6.** Double-materiality matrix for Port Esbjerg.

Combining with the REGS model, the double-materiality matrix identifies KPIs in all four quadrants: resilience, emissions, governance and sharing, as illustrated in Figure 7.

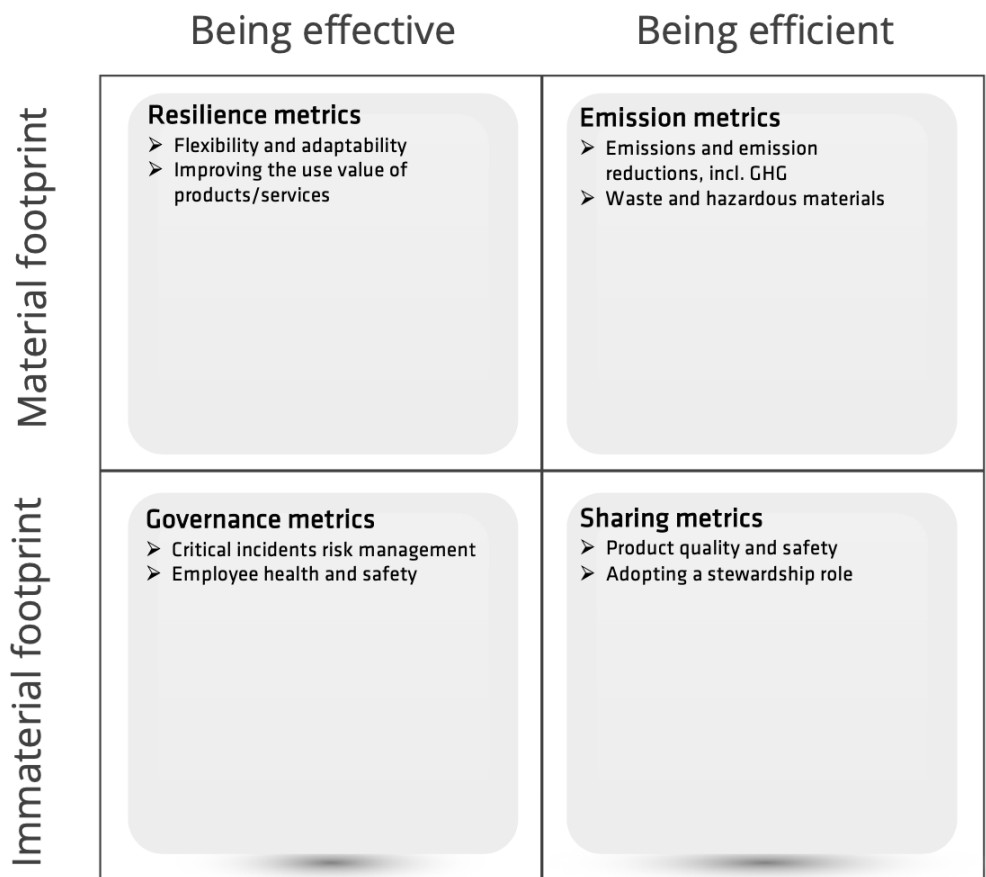

**Figure 7.** ESG KPIs are illustrated through the REGS model (adapted from [21]).

The ESG metrics identified in Figure 7 represent the KPIs that Port Esbjerg should focus on in their double-materiality explanation. The double-materiality assessment in the present case study used a business model analysis approach to identify relevant business model configurations for the company's customer segments. These business model configurations helped identify the value drivers and competitive parameters, which again formed the basis of understanding what the material impacts and footprints of this given organisation comprised. If this analysis had been performed on another port, the results would likely have differed due to different strategic focus, competencies, location and corporate culture, to name a few contingent factors.

## 6. Conclusions

Recent accounting regulations impose the perspective of double materiality upon sustainability reporting. This is to reconcile the financially oriented ESG perspective with the socially oriented CSR perspective. An orientation towards financial materiality drives the financially oriented ESG perspective. This means that investors, analysts, creditors and the like are focused on how climate risks will affect future Return on Assets and Investments, denoted as ROA and ROI. The socially oriented CSR perspective is concerned with doing good for the planet, people and society. Therefore, it comprises an impact materiality perspective.

Business models have been shown to hold the promise of increasing the transparency of companies and their value creation [29], especially when that value creation needs to consider aspects of sustainability [30] such as environmental, social or governance aspects. Therefore, using a business model as a natural connection between double materiality, ESG metrics and ESG reporting is a natural step [29]. Despite this, companies need to be aware of potential problems when prioritising what is material and what is not, what the

thresholds for disclosure should be and how such information should be disclosed. These aspects need more thorough research in the future. Another arena for future research is linking various ESG KPI databases, including those suggested by the ESRS, the SASB and Bloomberg.

This article contributes to the extant literature on double materiality by illustrating how business models can articulate the missing link between a value chain analysis and KPIs based on double-materiality assessments through a case study. Due to applying a single case study, readers should be aware of the limitations in generalising the findings when using such a methodological approach. The practical implications of this paper lie in the combinatorial application of existing strategy and business modelling methods to demystify and solve recent regulatory requirements with which many companies, including small- and medium-sized enterprises, are anxious about. In conclusion, the case illustrated in this paper also provides evidence that it is possible to turn recent regulatory requirements from accounting standard setters from being just another costly reporting exercise into a strategy-improvement routine that can inspire business model innovation and business opportunities for future resilience.

**Funding:** This research received external funding from the Port of Aalborg, Denmark.

**Institutional Review Board Statement:** Not applicable.

**Informed Consent Statement:** Not applicable.

**Data Availability Statement:** No new data were created or analyzed in this study. Data sharing is not applicable to this article.

**Acknowledgments:** The author wishes to thank the research assistants Sebastian Stück, Peter Thomsen and Brian Andersen for their work on data collection and initial analyses.

**Conflicts of Interest:** The author declares no conflict of interest.

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
