# Peer review of "ESG Reporting and Metrics: From Double Materiality to Key Performance Indicators"

_sustainability, doi:10.3390/su152416844_

Round 1

Reviewer 1 Report

Comments and Suggestions for Authors

Dear Author,

Your paper on "ESG Reporting and Metrics: From Double Materiality to Key Performance Indicators" showcases a commendable effort in exploring critical aspects of sustainable business practices. Here are some review comments and suggestions for revisiting the article:

1. Clarity of Contributions:

   - Clarify the specific contributions your paper makes concerning existing literature. A more explicit statement on the novel aspects your research introduces will enhance the paper's impact.

2. Enhancing Innovation:

   - Consider incorporating elements of cutting-edge technologies or interdisciplinary approaches to infuse further innovation into your exploration of ESG reporting and metrics.

3. Strengthening Quality:

   - Strengthen the academic rigor by integrating additional peer-reviewed references. Address potential counterarguments to fortify the robustness of your arguments and analysis.

4. Practical Implementations:

   - To enhance the practical applicability of your work, consider integrating more tangible examples or case studies. This will help readers bridge the gap between theory and real-world implementation.

5. Balancing Depth and Accessibility:

   - Maintain the depth of analysis while ensuring accessibility. Consider providing a brief summary or key takeaways at strategic points to aid readers in navigating complex concepts.

6. Engage with Emerging Trends:

   - Explore and discuss emerging trends or future directions in ESG reporting. This will add a forward-looking dimension to your paper and position it at the forefront of current discussions.

Overall, your work is promising, and these suggestions aim to further refine its impact and practical relevance. I look forward to seeing how you address these points in your revised manuscript.

Best regards

Comments on the Quality of English Language

The quality of English language in this article is generally strong. The writing is clear, and the ideas are presented in a coherent manner. However, attention to detail could enhance the overall polish of the language. Paying close consideration to grammar, punctuation, and sentence structure will elevate the article's professional tone. Additionally, ensuring consistency in terminology and refining word choices for precision will contribute to a more refined linguistic quality. Overall, with some meticulous editing, the article's English language can be further improved.

Author Response

Reviewer response:   Dear Author,   Your paper on "ESG Reporting and Metrics: From Double Materiality to Key Performance Indicators" showcases a commendable effort in exploring critical aspects of sustainable business practices. Here are some review comments and suggestions for revisiting the article:   1. Clarity of Contributions: - Clarify the specific contributions your paper makes concerning existing literature. A more explicit statement on the novel aspects your research introduces will enhance the paper's impact.   In accordance with the comments from two other reviewers, clarity has now been added by including an explicit research question as well as reformulations .   2. Enhancing Innovation: - Consider incorporating elements of cutting-edge technologies or interdisciplinary approaches to infuse further innovation into your exploration of ESG reporting and metrics.   Thank you for this comment. However, cutting-edge technologies, such as 6G, AI abd Blockchain are irrelevant in this context. This subject is highly interdisciplinary because it combines the need for financial materiality from investors and auditors with the needs from society to have a much stronger focus on reducing corporate footprints.   3. Strengthening Quality: - Strengthen the academic rigor by integrating additional peer-reviewed references. Address potential counterarguments to fortify the robustness of your arguments and analysis.   Thank you. This has been improved as much as possible. However, because this perspective is so young there is not much academic research out there to cite yet.   4. Practical Implementations: - To enhance the practical applicability of your work, consider integrating more tangible examples or case studies. This will help readers bridge the gap between theory and real-world implementation.   I believe that the case study offered does exactly this. This is also a key strength of the current paper and a reason it is expected to get many citations in the future. The case description has been rewritten so that it is clearer to the reader.   5. Balancing Depth and Accessibility: - Maintain the depth of analysis while ensuring accessibility. Consider providing a brief summary or key takeaways at strategic points to aid readers in navigating complex concepts.   In accordance with this comment and the issues raised by other reviewers, the abstract, introduction and conclusion have now been improved to do this.   6. Engage with Emerging Trends: - Explore and discuss emerging trends or future directions in ESG reporting. This will add a forward-looking dimension to your paper and position it at the forefront of current discussions.   I believe this is actually exactly what is done, considering the fact that the regulations and processes have been released in the last 3-4 months.   Overall, your work is promising, and these suggestions aim to further refine its impact and practical relevance. I look forward to seeing how you address these points in your revised manuscript.   Best regards   Comments on the Quality of English Language The quality of English language in this article is generally strong. The writing is clear, and the ideas are presented in a coherent manner. However, attention to detail could enhance the overall polish of the language. Paying close consideration to grammar, punctuation, and sentence structure will elevate the article's professional tone. Additionally, ensuring consistency in terminology and refining word choices for precision will contribute to a more refined linguistic quality. Overall, with some meticulous editing, the article's English language can be further improved.   The manuscript has been edited for grammar and proofread.

Reviewer 2 Report

Comments and Suggestions for Authors

I recommend the author offer more details in the Research Design and Methods section, as no concrete data/information can be found, especially about the semi-structured interviews and the data collection exercise. What kind of data was collected, how was the data analyzed, and how did this analysis finally drive the results, discussions, and conclusions. If there are limitations to your research please discuss them in the Conclusions section

Author Response

Reviewer 1 I recommend the author offer more details in the Research Design and Methods section, as no concrete data/information can be found, especially about the semi-structured interviews and the data collection exercise. What kind of data was collected, how was the data analyzed, and how did this analysis finally drive the results, discussions, and conclusions. If there are limitations to your research please discuss them in the Conclusions section   Thank you for these comments, which are most helpful.   The research design has now been improved to include information on the interview questions and how the information was analysed. As suggested by the reviewer, limitations to using case studies for generalisation purposes has now been added to the conclusions.

Reviewer 3 Report

Comments and Suggestions for Authors

The issue presented in the study raises relevant topics related to the connections between firms’ value chains and the requirements for double materiality assessments in contemporary reporting rules. The topic is actual and of interest. The title is clear and appropriate to the paper’s subject matter.

The manuscript has a theoretical-empirical character. Based on theoretical and in-depth empirical research, aspects of the connection between the due diligence analysis of firms’ value chains and distribution networks, the analysis of the competitive parameters of the business model, the identified impacts, risks, and opportunities, and the double materiality perspective are presented.

The text is written clearly, concisely, stylistically, and technically correct. Actual scientific material, logically and reasonably presented. The literature used makes it possible to reveal the degree of knowledge of the problem and highlight issues that require further study.  Every reference cited in the text is also present in the reference list.

When developing the article, a sufficient number of literary and informational sources, amounting to 35 in total, were used, as a confirmation of a very good literary awareness of the authors. Part of the interpreted data is illustrated with the help of 8 figures.

Below are several suggestions that I hope will be helpful in the paper, at the author's discretion:

1)      in terms of structure, at the discretion of the authors, it is possible within the exposition, after the introduction, to set aside an independent paragraph highlighting the methods and the methodological framework;

2)      the relationship between topics and aspects of topics presented in Table 1 should be specified and put in order;

3)      the authors’ statement  “These ESG metrics represent precisely the objective of the present article, namely, to identify KPIs based on a double materiality assessment”  requires additional justification and clarification (lines 442-443);

4)      in the Conclusions section, it would be helpful to highlight issues that require further study.

Author Response

Reviewer 2 The issue presented in the study raises relevant topics related to the connections between firms’ value chains and the requirements for double materiality assessments in contemporary reporting rules. The topic is actual and of interest. The title is clear and appropriate to the paper’s subject matter. The manuscript has a theoretical-empirical character. Based on theoretical and in-depth empirical research, aspects of the connection between the due diligence analysis of firms’ value chains and distribution networks, the analysis of the competitive parameters of the business model, the identified impacts, risks, and opportunities, and the double materiality perspective are presented.   The text is written clearly, concisely, stylistically, and technically correct. Actual scientific material, logically and reasonably presented. The literature used makes it possible to reveal the degree of knowledge of the problem and highlight issues that require further study.  Every reference cited in the text is also present in the reference list.  When developing the article, a sufficient number of literary and informational sources, amounting to 35 in total, were used, as a confirmation of a very good literary awareness of the authors. Part of the interpreted data is illustrated with the help of 8 figures. Below are several suggestions that I hope will be helpful in the paper, at the author's discretion: 1)      in terms of structure, at the discretion of the authors, it is possible within the exposition, after the introduction, to set aside an independent paragraph highlighting the methods and the methodological framework;   The methodology is of course explained in section 4, however, a short clause has been added in the introduction as suggested by the reviewer.   2)      the relationship between topics and aspects of topics presented in Table 1 should be specified and put in order;   Thank you for this comment. This has now been changed and the table has been removed in accordance with other comments. The order followed is the order that the original publication applies.   3)      the authors’ statement  “These ESG metrics represent precisely the objective of the present article, namely, to identify KPIs based on a double materiality assessment”  requires additional justification and clarification (lines 442-443);   Thank you for this comment. The text has now been amended accordingly. It was not clear what was meant, but now it should be clearer that these are the KPIs the analysis identified.   4)      in the Conclusions section, it would be helpful to highlight issues that require further study.   Thank you for this comment. The conclusion did briefly mention future research on thresholds. In addition to this, the conclusion now also mentions future research into the links between ESG KPIs and ESRS, SASB and Bloomberg KPIs.

Reviewer 4 Report

Comments and Suggestions for Authors

Dear author,

Thank you so much  for the opportunity to read this interesting project.

Good Luck.

Best Wishes,

The Reviewer

Comments on the Quality of English Language

Professional proofreading is recommended.

Author Response

Reviewer 3 Thank you for the possibility to read and discuss this interesting work dealing with double materiality. All comments are suggestive to improve the quality of the paper from readers` perspective. However, the following points could be advantageous. - It will be more clearer to the reader if the author/s could add this research results implication within the abstract;   Thank you for this comment. The latter part of the abstract has been reformulated to accommodate this suggestion   - Page 2 point 1. The research contribution is completely excluded;   This is indeed a valid point. The following has been added: The present article answers the research question; How can firms create a link be-tween their business model and the ESG KPIs that document their double materiality statement?   - Page 3 point 2. Clear definition of double materiality is essential but omitted from this study;   This is a valid point, and on lines 121-123, the following is stated: Double materiality means that the company has to evaluate its footprint on the environment and society on the one hand and how environmental and societal factors influence the organisation on the other.   - It will be more clearer to the readers if the author/s compare this study`s results with prior studies. Justify why if this study result is than the prior studies;   Unfortunately, there are no prior studies that look at the connections between double materiality and KPIs. This justifies the reason for publishing the case.   - The limitations of the study and recommendations for future researcher are very crucial point but both of them are clearly excluded within this study.   Yes, this is a great comment and this has now been included in the methodology section and the conclusion section

Reviewer 5 Report

Comments and Suggestions for Authors

The article presents a case, while it is mostly a theoretical article. The structure of the article is correct. It is a scientific study. 

To increase the scientific quality, one should: 

 - a research question should be posed - this will help for more targeted consideration

- article should refer to the research literature - conduct a more thorough review of the lietrature. The author did a very good job of reviewing the practical aspects (mainly ineternet sites), but there is a lack of showing what is happening in the literature

Author Response

Reviewer 4 The article presents a case, while it is mostly a theoretical article. The structure of the article is correct. It is a scientific study.   Thank you for this kind comment.   To increase the scientific quality, one should:   - a research question should be posed - this will help for more targeted consideration   Thank you for this comment, it is most helpful. A research question "How can firms create a link between their business model and the ESG KPIs that document their double materiality statement?", has been added, line 57-59.   - article should refer to the research literature - conduct a more thorough review of the lietrature. The author did a very good job of reviewing the practical aspects (mainly ineternet sites), but there is a lack of showing what is happening in the literature   I am not quite sure I understand this comment. This article does not use Internet sites, but refers mostly to academic literature and in addition a few reports from for example the OECD, European Commission etc.

Round 2

Reviewer 1 Report

Comments and Suggestions for Authors

I appreciate your diligent efforts in addressing my suggestions. The improvements in the revised manuscript are noteworthy. The enhanced structure and improved logical flow contribute significantly to the overall coherence of the narrative. Furthermore, your refined language expression demonstrates increased precision and clarity. Overall, your prompt response to feedback and the substantial enhancements made in the manuscript are commendable, further strengthening the academic contribution.

Author Response

Thank you for your most helpful comments that helped me improve the article. I really appreciate it. I have made a careful note to take your recommendations into consideration. 

Reviewer 2 Report

Comments and Suggestions for Authors

The manuscript can be accepted in this form

Author Response

Thank you for your most helpful comments that have helped me to improve the article. It is much appreciated.